# Comparative Protective Efficacies of Novel Avian Paramyxovirus-Vectored Vaccines against Virulent Infectious Bronchitis Virus in Chickens

**DOI:** 10.3390/v12070697

**Published:** 2020-06-28

**Authors:** Edris Shirvani, Siba K. Samal

**Affiliations:** Virginia-Maryland College of Veterinary Medicine, University of Maryland, College Park, MD 20742, USA; eshirvan@umd.edu

**Keywords:** infectious bronchitis virus, avian paramyxovirus vaccine vector, vaccine vector replication, poultry vaccines

## Abstract

Viral vectored vaccines are desirable alternatives for conventional infectious bronchitis virus (IBV) vaccines. We have recently shown that a recombinant Newcastle disease virus (rNDV) strain LaSota expressing the spike (S) protein of IBV strain Mass-41 (rLaSota/IBV-S) was a promising vaccine candidate for IBV. Here we evaluated a novel chimeric rNDV/avian paramyxovirus serotype 2 (rNDV/APMV-2) as a vaccine vector against IBV. The rNDV/APMV-2 vector was chosen because it is much safer than the rNDV strain LaSota vector, particularly for young chicks and chicken embryos. In order to determine the effectiveness of this vector, a recombinant rNDV/APMV-2 expressing the S protein of IBV strain Mass-41 (rNDV/APMV-2/IBV-S) was constructed. The protective efficacy of this vector vaccine was compared to that of the rNDV vector vaccine. In one study, groups of one-day-old specific-pathogenic-free (SPF) chickens were immunized with rLaSota/IBV-S and rNDV/APMV-2/IBV-S and challenged four weeks later with the homologous highly virulent IBV strain Mass-41. In another study, groups of broiler chickens were single (at day one or three weeks of age) or prime-boost (prime at day one and boost at three weeks of age) immunized with rLaSota/IBV-S and/or rNDV-APMV-2/IBV-S. At weeks six of age, chickens were challenged with a highly virulent IBV strain Mass-41. Our challenge study showed that novel rNDV/APMV-2/IBV-S provided similar protection as rLaSota/IBV-S in SPF chickens. However, compared to prime-boost immunization of chickens with chimeric rNDV/APMV-2, rLaSota/IBV-S and/or a live IBV vaccine, single immunization of chickens with rLaSota/IBV-S, or live IBV vaccine provided better protection against IBV. In conclusion, we have developed the novel rNDV/APMV-2 vector expressing S protein of IBV that can be a safer vaccine against IB in chickens. Our results also suggest a single immunization with a LaSota vectored IBV vaccine candidate provides better protection than prime-boost immunization regimens.

## 1. Introduction

Infectious bronchitis (IB) is a highly contagious viral disease of poultry [1]. It is predominantly a respiratory disease, but it can also affect renal and reproductive tracts. Respiratory distress, poor carcass weight, substandard egg quality, and decreased egg production resulting from IB cause huge economic losses for the poultry industry worldwide [1,2].

Infectious bronchitis virus (IBV) is a member of the family *Coronaviridae.* The genome of IBV is a non-segmented, positive-strand RNA of about 27.6 kilobase (kb) in length. The genome consists of five basic and four accessory genes in the order 5′-leader-replicase-UTR-S(spike)-3a,3b-M(membrane)-E(envelope)-5a,5b-N(neuclocapsid)-UTR-polyA-3′. IBV is an enveloped virus and contains S, M, E, and N proteins [3,4].

Vaccination is the most efficacious approach to control IB. Currently, both inactivated and live attenuated vaccines are used to control IB [1]. However, inactivated vaccines do not elicit strong mucosal and cell mediated immunity [5,6]. The production cost and application methods are other limiting factors for inactivated vaccines [7,8]. The live-attenuated IBV vaccines also have several disadvantages: i) live attenuated IBV vaccines can revert to their original virulence after circulation in the field [7,8]; ii) they can create new variant viruses through recombination and mutations [9,10]; iii) they can cause pathological signs in immunized young birds [11,12]; and iv) the protective efficacy of live attenuated vaccines is also not guaranteed [13]. Therefore, the development of a novel safe, efficacious, and affordable vaccine against IBV is necessary. 

Viral-vectored vaccines provide an alternative approach for current IBV vaccines. Although several viral vectors have been used to develop novel IBV vaccines, each viral vector has its own advantages and disadvantages [14,15,16,17,18]. Among the viral vectors used for IBV vaccines, avian paramyxovirus (APMV) vectors are more attractive because both APMV and IBV infect chickens through the respiratory tract. Therefore, an APMV vector IBV vaccine will elicit a strong local antibody response, in addition to a systemic response. Furthermore, a single vaccine can be used to protect two poultry pathogens. APMV vector vaccines are highly safe, which can overcome some disadvantages associated with live-attenuated IBV vaccines [19]. Avian paramyxoviruses (APMVs) belong to the family *Paramyxoviridae* and have a non-segmented, negative-sense RNA genome that contains six genes in the order 3′-N-P-M-F-H-L-5′ [20]. To date, twenty serotypes (APMV-1 through APMV-20) have been officially reported [21]. APMV-1 strains are classified into velogenic (highly virulent), mesogenic (moderately virulent), and lentogenic (avirulent) pathotypes based on their pathogenicity in chickens. Velogenic and mesogenic strains cause Newcastle disease (ND) and are called Newcastle disease virus (NDV). Lentogenic APMV-1 strains have been successfully used as safe live vaccines against ND for over 60 years [22]. These strains have also been used as vaccine vectors against several avian and non-avian pathogens [23]. 

We have recently shown that a recombinant lentogenic APMV-1 strain LaSota expressing the S protein of IBV was a promising vaccine candidate against IBV [19]. However, APMV-1 strain LaSota can induce mild clinical signs in young chicks, and it is not safe for chicken embryos. Therefore, APMV serotypes other than APMV-1 have been investigated as vaccines vectors [24,25,26]. Among other APMV serotypes, APMV-2 is an alternative vaccine vector for chickens because it replicates well in chickens, rarely causes clinical signs in chickens, and it is highly safe for chicken embryos [27,28]. APMV-2 replication is restricted to the respiratory tract, therefore, it has the advantage of not leaking into environment [26]. Intracerebral inoculation of one-day-old chicks showed that APMV-2 neither caused disease nor replicated detectably in brain tissue [28]. APMV-2 grows to similar level as APMV-1 in embryonated chicken eggs, but it is safer than APMV-1 for chicken embryos [28]. APMV-2 also has the least serological cross-reaction with APMV-1 [29]. It is not hampered by pre-existing immunity to NDV [26]. These features make APMV-2 a suitable vaccine vector for chickens. Unfortunately, it was found that addition of a foreign gene highly debilitates the replication of APMV-2, making it unsuitable for use as a vaccine vector (our unpublished data). To solve this obstacle, a novel recombinant chimeric vector in which the ectodomains of fusion (F) and hemagglutinin neuraminidase (HN) proteins of NDV were replaced with the corresponding ectodomains of F and HN proteins of APMV serotype 2 (rNDV/APMV-2) was developed in our laboratory [30]. It was shown that F and HN are major contributors in replication, tropism, and pathogenicity of APMVs [31]. The chimeric virus was highly safe for chickens and chicken embryo and replicated efficiently in vivo [31]. The F and HN proteins of APMV-1 and APMV-2 have only 41% and 35% amino acid sequence identity, respectively [28]. The seroconversion rates to APMV-1 and APMV-2 in commercial poultry are 71% and 15%, respectively [32]. These characteristics also make rNDV/APMV-2 a highly safe vector for chickens that has the potential for *in ovo* immunization and administration in presence of pre-existing immunity to APMV-1. 

In this study, we have evaluated the chimeric rNDV/APMV-2 vector virus as a vaccine vector against IBV in comparison with the rAPMV-1 strain LaSota (rLaSota) vector and a modified version of the rAPMV-1 strain LaSota (rLaSota-527) vector, with increased replication [33]. We generated rNDV/APMV-2 and rLaSota-527 expressing the full-length S protein of IBV strain Mass-41. We have evaluated the protective efficacies of rNDV/APMV-2, rLaSota and rLaSota-527 vectors expressing S protein of IBV in chickens against the homologous highly virulent IBV strain Mass-41 challenges. Our results showed that immunization of chicks with rNDV/APMV-2 or rLaSota expressing S protein provided comparable protection against IBV challenge. However, we found that compared to prime-boost immunization regimens using rNDV/APMV-2 expressing S protein, rLaSota expressing S protein and/or live IBV vaccine, single immunization of chickens with rLaSota expressing S protein, or live attenuated IBV vaccine provided better protection against IBV challenge. Furthermore, we showed that compared to rLaSota expressing S protein, immunization of chickens with rLaSota-527 expressing S protein, with increased replication, provided less protection against IBV challenge. We have developed a vaccine candidate for IBV using a novel chimeric rNDV/APMV-2 vector that is safer for chickens. 

## 2. Materials and Methods

### 2.1. Cells and Viruses

Human epidermoid carcinoma (HEp-2) cells and chicken embryo fibroblast (DF1) cells were obtained from the American Type Culture Collection (ATCC; Manassas, VA) to recover recombinant viruses by reverse genetics and for in vitro characterization of recovered viruses, respectively. The cells were grown in Dulbecco’s minimal essential medium (DMEM) containing 10% fetal bovine serum (FBS). Recombinant viruses were propagated in allantoic fluids of 10-day-old embryonated specific pathogen free (SPF) chicken eggs (Charles Rivers, MA). The virulent IBV strain Mass-41 was propagated in 10-day-old embryonated SPF chicken eggs. The allantoic fluid containing IBV was harvested five days after infection, centrifuged at low speed to sediment debris and the supernatants were aliquoted and stored at −70 °C. The titer of the stored virus was determined by 50% embryo infectious dose (EID_50_). This virus stock was used to infect chickens in IBV challenge experiments.

### 2.2. Generation and Growth Characteristics of rNDV/APMV-2 and rLaSota-527 Expressing S Protein of IBV 

A plasmid BR322 containing full anti-genomic cDNA of NDV strain Beaudette C (BC) in which the F and HN ectodomains of NDV were replaced with corresponded F and HN ectodomain of APMV-2 (GenBank Accession No. EU338414) was constructed previously [30]. A plasmid containing full anti-genomic cDNA of APMV-1 strain LaSota (GenBank Accession No. AF077761.1) with a Y527A mutation in cytoplasmic tail of F protein also was constructed previously [33]. Here we digested a transcription cassette containing the full-length S gene of IBV strain Mass-41 (GenBank Accession No. AY851295.1) from the pBR322 containing full anti-genomic cDNA of NDV strain LaSota expressing S gene of IBV [19] using the *PmeI* enzyme and inserted between P and M genes of NDV/APMV-2 and LaSota-527 using the *PmeI* restriction enzyme site. The transcription cassette contained the following sequences in a 3′ to 5′ order; *PmeI* restriction enzyme site, 15 nucleotides of NDV UTR, gene end (GE) of phosphoprotein (P) gene of NDV, one T nucleotide as intergenic sequence (IGS), gene start (GS), Kozak, open reading frame (ORF) of the S gene of IBV and *PmeI* restriction enzyme site. The ORF of S gene of IBV strain Mass-41 was codon optimized for higher expression in chicken cells (GenScript; optimization on Gallus Gallus codons using OptimumGene TM PSO algorithm). The correct sequence of flanking region was confirmed by sequence analysis using a forward primer from the P gene of NDV at upstream of the *PmeI* restriction enzyme site and a reverse primer from the M gene of NDV at downstream of the *PmeI* restriction enzyme site. The rNDV-APMV-2 or rLaSota-527 expressing S protein of IBV were recovered from cDNAs using reverse genetics as described previously [34]. The recovered rNDV-APMV-2 expressing S protein of IBV and rLaSota-527 expressing S protein of IBV were named rNDV/APMV-2/IBV-S and rLaSota-527/IBV-S, respectively. The rNDV-APMV-2/IBV-S and rLaSota-527/IBV-S were passaged in 10-day-old embryonated eggs. The viruses with high HA titers were harvested and stored at −70 °C in vials of aliquots. A rLaSota expressing S protein of IBV (rLaSota/IBV-S), which was generated previously, also was used in this study [19]. In two separate experiments, the plaque morphology of rNDV-APMV-2/IBV-S or rLaSota-527/IBV-S in comparison with rLaSota/IBV-S were assayed in DF1 cells under DMEM containing 0.8% methylcellulose and 10% fresh allantoic fluid over layer. 

### 2.3. Expression of S Protein by rNDV-APMV-2/IBV-S and rLaSota-527/IBV-S

The expression of S protein by rNDV-APMV-2/IBV-S and rLaSota-527/IBV-S in comparison with the expression of S protein by rLaSota/IBV-S was assessed by Western blot. The monolayer of DF1 cells was infected with rLaSota, rLaSota/IBV-S, rLaSota-527/IBV-S or rNDV-APMV-2/IBV-S in the presence of 10% allantoic fluid. The DF1 cell lysates at almost similar level of infections were collected 30 h after infection. The expression of S protein in cell lysates was detected by Western blot analysis using a chicken polyclonal anti IBV serum.

### 2.4. The Protective Efficacies of rNDV-APMV-2/IBV-S and rLaSota/IBV-S Expressing S Protein against IBV Challenge in SPF Chickens

A total of 34 one-day-old SPF chicks obtained from Charles Rivers (Charles Rivers MA) were divided into three groups (Groups 1–3) of ten chicks and one group of four chicks (Group 4). The chicks of Groups 1 and 2 were immunized with 10^6^ EID_50_/bird, in 200 µl volume, of rNDV-APMV-2/IBV-S and rLaSota/IBV-S through oculonasal route, respectively. The chicks of Group 3 were inoculated with 200 µl PBS through oculonasal route, and chicks of Group 4 were left uninfected. Four weeks after immunization, all birds of groups one to three were infected with 10^4^ EID_50_/bird, in 200 µl volume, of virulent IBV strain Mass-41 through eye drop route and birds of group four were left uninfected. The severity scores of clinical signs of IBV including, nasal discharge, ocular discharge, and difficulty in breathing (0 = normal; 1 = presence of mild ocular discharge, mild nasal discharge, and/or sneezing; 2 = presence of heavy ocular discharge and/or heavy nasal discharge with mild tracheal rales and mouth breathing, and/or coughing; 3 = heavy ocular discharge and heavy nasal discharge with sever tracheal rales and mouth breathing, gasping, dyspnea, and/or severe respiratory distress) were recorded twice a day for 10 days post-challenge. At day five post-challenge, tracheal swab samples were collected from all chickens and placed into 3 mL cold DMEM containing 10X antibiotics. The viral RNA was quantified in swab samples by an IBV-N gene-specific RT-qPCR using the protocol described previously [19].

### 2.5. The Protective Efficacies of rNDV-APMV-2 and rLaSota Expressing S Protein against IBV Challenge in Broiler Chickens

A total of 48 one-day-old broiler chickens were divided into eight groups, six in each. In a single or prime-boost immunization (prime at day one and boost at week three of age), chicks were immunized with 2^7^ HAU/bird, in 200 μL volume of rNDV-APMV-2/IBV-S, rLaSota/IBV-S expressing S protein, and/or recommended dose 200 μL in volume of a live attenuated Mass-type IBV vaccine by the oculonasal route based on the immunization regimen for each group listed in Table 1. At week six of age (six weeks-post prime immunization and three weeks post-boosting), birds were infected with 10^5.5^ EID_50_/bird of virulent IBV strain Mass-41 by the oculonasal route. The severity scores of clinical signs of IBV were recorded and post-challenge tracheal viral load were evaluated as described for IBV challenge experiment in SPF chickens. 

All animal experiments were performed in our USDA approved Biosafety level-2 and Biosafety level-2 plus facilities following the guidelines and approval of the Institutional of Animal Care and Use Committee (IACUC), University of Maryland.

### 2.6. Statistical Analysis

Data were analyzed among groups using a one-way-ANOVA test. The *t*-test was used to compare two groups. To avoid bias, IBV protective experiments were designed as blinded studies.

## 3. Results

### 3.1. Generation and Growth Characterization of rNDV/APMV-2 or rLaSota-527 Expressing S Protein of IBV

The transcription cassette containing the S gene of IBV was inserted between the P and M genes in two individual plasmids containing cDNA of the full length antigenomic RNA of NDV-APMV-2 or LaSota-527. rNDV-APMV-2 expressing IBV S protein (rNDV-APMV-2/IBV-S) and rLaSota-527 expressing IBV S protein (rLaSota-527/IBV-S) were recovered from cDNAs successfully. A rLaSota expressing S protein of IBV (rLaSota/IBV-S) which we generated previously also was used in the in vitro and in vivo experiments of this study [19]. The recovered viruses replicated in 10-day-old embryonated SPF chicken eggs efficiently. The plaque morphology assay showed that either rNDV-APMV-2 or rNDV-APMV-2/IBV-S did not form visible plaques in DF1 cells under DMEM containing 0.8% methyl cellulose over layer (Figure 1C). Compared to rLaSota/IBV-S, rLaSota-527/IBV-S formed slightly larger plaques in DF1 cells (Figure 1D).

### 3.2. Expression of the S Protein of IBV by rNDV-APMV-2/IBV-S, rLaSota-527/IBV-S and rLaSota/IBV-S

The expression of S protein of IBV by rNDV-APMV-2/IBV-S, rLaSota-527/IBV-S, and rLaSota/IBV-S was detected by Western blot. Like rLaSota/IBV-S, rNDV-APMV-2/IBV-S and rLaSota-527/IBV-S expressed the S protein at high level (Figure 2). For the S protein of IBV expressed from rNDV-APMV-2/IBV-S (Figure 2 lane 1), rLaSota-527/IBV-S (Figure 2 lane 2), and rLaSota/IBV-S (Figure 2 lane3), the bands on top (~170–220 kDa) represent either uncleaved S protein (S0) or polymeric forms of S protein. The ~130 kDa, the ~95 kDa, and the ~60 kDa bands represent S1 or S2 subunit of cleaved S protein of IBV.

### 3.3. The Protective Efficacies of rNDV-APMV-2/IBV-S and rLaSota/IBV-S against IBV Challenge in SPF Chickens

In order to evaluate the comparative protective efficacies of rNDV-APMV-2/IBV-S and rLaSota/IBV-S against IBV, one-day-old SPF chicks were immunized with 10^6^ EID_50_/bird of rNDV-APMV-2/IBV-S or rLaSota/IBV-S and four weeks post-vaccination they were challenged with 10^4^EID_50_/bird.

Compared to non-immunized chicks, both rNDV-APMV-2/IBV-S and rLaSota/IBV-S provided significant and comparable protection against clinical signs of IBV (Figure 3A). However, neither rNDV-APMV-2/IBV-S nor rLaSota/IBV-S decreased the post-challenge viral load in trachea (Figure 3B). The antibodies induced against LaSota and rNDV-APMV-2 were assessed by hemagglutination inhibition (HI) assay using the protocol of OIE [22] rNDV-APMV-2/IBV-S and rLaSota/IBV-S induced antibodies against rNDV-APMV-2 and rLaSota, respectively; however, compared to rNDV-APMV-2/IBV-S, rLaSota/IBV-S induced slightly higher HI titers (Figure 3C). 

### 3.4. The Protective Efficacies of rNDV-APMV-2/IBV-S and rLaSota/IBV-S against IBV Challenge in Broiler Chickens

Eight groups of broiler chickens were immunized with single or prime-boost immunization regimens listed in Table 1. At week six of age, chickens were infected with 10^5.5^ EID_50_/bird of virulent IBV. The results showed that immunization of chickens with rLaSota/IBV-S, rNDV-APMV-2/IBV-S, or live IBV vaccine, no matter which regimens of immunization, reduced the post-challenge IBV clinical signs significantly. However, compared to other groups, immunization of broiler chickens with rLaSota/IBV-S or live IBV vaccine at week three of age provided better protection of clinical signs of IBV. Single immunization of chickens with rLaSota/IBV-S or live IBV vaccine at week three of age reduced post-challenge tracheal viral load significantly, whereas single immunization of chickens at day one of age and prime-boost immunization regimens did not decreased post-challenge viral load (Figure 4B). Antibody titers against rNDV/APMV-2 and/or rLaSota were detected in corresponded immunized groups of chickens by HI assay using the protocol of OIE (Figure 5) [22].

We also compared protective efficacies of rLaSota-527/IBV-S and rLaSota/IBV-S for protecting against clinical signs and in preventing post-challenge viral shedding against IBV challenge. Four-week-old chickens were immunized and three weeks after immunization, they were infected with virulent IBV strain Mass-41. The clinical signs of IBV were recorded, and post-challenge tracheal IBV RNA load was quantified by RT-qPCR. Three weeks after immunization the humoral antibodies induced against LaSota were detected by the HI assay. Our results showed that compared to rLaSota-527/IBV-S, rLaSota/IBV-S provided slightly better protection against clinical signs. Compared to rLaSota-527/IBV-S, the reduction in post-challenge tracheal viral load for rLaSota/IBV-S was 5 Log_10_ higher. However, our result showed that compared to rLaSota/IBV-S, immunization of chickens with rLaSota-527/IBV-S induced a higher level of HI titers of antibodies to NDV. We did not show the figure here, because parts of data (data for the protective efficacy of rLaSota/IBV-S) were presented in Figure 5 of our previous publication [19]. 

## 4. Discussion

The control of IB is achieved by vaccination [1]. However, the currently used IBV vaccines are not safe and efficacious (5-13). Viral vectored vaccines are desirable alternatives for current IBV vaccines. Several viral vectors have been investigated to develop vaccines against IBV (14-18). Among these vectors NDV lentogenic strains have shown promising results as vaccine vectors in protecting chickens against IBV [19]. The immunization of chicks at an early age against IBV is essential. Here, we have evaluated an alternative chimeric APMV vector in which the internal proteins are of NDV, but the surface proteins are from APMV-2 [30]. This chimeric vector is highly safe for chicken embryos and young chicks [30,31]. Our results indicate that this novel chimeric vector is comparable to rNDV strain LaSota vector in protecting chickens against IBV. However, because of low pathogenicity of rNDV/APMV-2 vector [30,31], rNDV/APMV-2/IBV-S is a better choice for vaccination of chickens and may be applicable for *in ovo* vaccination. 

Our result showed that rNDV/APMV-2/IBV-S did not develop plaques in DF1 cells, whereas, rLaSota/IBV-S and rLaSota-527/IBV-S formed a large size plaques. This result was not unexpected, because APMV-2 does not form syncytia or plaques in DF1 cells [27,31]. A previous study has also shown that compared to rLaSota, rLaSota-527 formed a larger size plaque in DF1 cells [33]. Similar to rLaSota/IBV-S, both rNDV-APMV-2/IBV-S and rLaSota-527/IBV-S expressed S protein at high levels suggesting that the full-length S protein of IBV strain Mass-41 is expressed efficiently by these APMV vectors. 

Immunization of SPF chicks with rNDV-APMV-2/IBV-S and rLaSota/IBV-S provided comparable levels of protection against IBV clinical signs. However, in this IBV protective experiment, neither rLaSota/IBV-S nor rNDV-APMV-2/IBV-S decreased the post-challenge viral load, significantly. One possible explanation for this is that chicks at day one of age have an immature immune system when vaccinated for IBV [35,36,37,38,39,40] resulting in induction of lower levels of protective immunity including lower avidity of IgG and IgA antibodies to IBV [41,42]. Therefore, it is possible that our vaccination at day one of age may have also induced lower levels of protective immunity, which was not sufficient to cease challenge virus replication in the respiratory tract. This could also be due to the other factors which may affect the outcomes of an IBV challenge experiment [13], because previously we showed that immunization of one-day-old chicks with rLaSota/IBV-S decreased post-challenge tracheal viral load, significantly [19]. 

It was shown by Cavangah et al. that amino acids within hypervariable region (S1 subunit) of IBV spike proteins are responsible for major neutralizing epitopes [43]. We previously showed that although S1 subunit contains major neutralizing epitopes, the interaction of S1 and S2 subunits in full-length S proteins is required for expressing the neutralizing epitopes in native conformation [19]. A similar observation was also reported for SARS-CoV-1 [44]. Therefore, immunization of chickens with rLaSota vector or chimeric rNDV-APMV-2 vector expressing the full-length S protein without any amino acid change in the S protein is necessary to protect chickens against IBV strain Mass-41 challenge, which causes sever disease in chickens, in consistent with our previous study [19]. Recently, Zegpi et al. developed a rLaSota expressing an artificially made trimeric ectodomain of S protein of IBV using a GCN4 trimerization motif at C-terminal, adding a heterologous signal sequence (CD5) at N-terminal and removing the cleavage motifs of S protein. Their vaccine candidate provided limited protection against less virulent IBV strain Arkansas [45]. We think that in their study the modifications made in the S protein may have changed the native conformation of the S protein leading to the loss of critical neutralizing epitopes. Our results also showed that compared to rLaSota/IBV-S, rNDV-APMV-2/IBV-S induced slightly lower HI titers against the corresponding backbone virus in immunized SPF chickens. This is consistent with previous studies that have shown that APMV-2 or rNDV/APMV-2 replicated less efficiently than LaSota [27,28,30,31].

Similar to the live IBV vaccine, single immunization of broilers at three weeks of age with rLaSota/IBV-S significantly reduced post-challenge IBV genome in the trachea of chickens; however prime-boost immunization (heterologous or homologous regimens) with rLaSota/IBV-S and/or rNDV-APMV-2/IBV-S did not reduce post-challenge tracheal IBV RNA load at six weeks of age. These results suggest that not only double vaccination with rAPMVs/IBV-S did not improve the challenge outcomes, but, surprisingly, it increased the post-challenge clinical signs and tracheal viral load. We also observed that compared to single immunization, prime-boost immunization of SPF chickens with a live IBV vaccine provided less protection (unpublished). This could be due to the high titer of antibodies induced against non-neutralizing epitopes within the S protein of IBV and/or due to the negative effect of the over infection with vaccine strains on the development of balanced immune responses against IBV. Previously, it was shown that coronavirus infections may cause antibody dependent enhancement (ADE) [46]. A study has also shown that the spike protein of SARS-CoV-1 triggers ADE [47]. 

Single immunization of broiler chickens with rNDV-APMV-2/IBV-S or rLaSota/IBV-S at day one of age or at week three of age provided protection against a high dose of IBV challenge. However, compared to single immunization at day one of age, immunization of three-week-old broilers with rLaSota/IBV-S provided better outcomes, in protecting from disease and preventing viral shedding. These results suggest that consistent with our previous study [19] and a recent study [33], compared to young chicks, immunization of adult chickens with rLaSota/IBV-S provided better protection. This could be related to the maturity of immune system and/or the shorter interval between immunization and challenge [13]. 

Furthermore, our results showed that compared to rLaSota/IBV-S, immunization of SPF chickens with rLaSota-527/IBV-S, did not provide better protection, indicating that increased replication of the vector may not necessarily lead to increased protection.

In conclusion, in this study we have shown that the novel chimeric rNDV/APMV-2 vector expressing S protein of IBV, which is a better candidate for IBV vaccination because it is safer for young chicks and chicken embryos, provided protection against IBV in chickens. Our results also suggest a single immunization with an APMV vectored IBV vaccine candidate, with optimal replication, provides better protection than prime-boost vaccination against IBV.

## Figures and Tables

**Figure 1 viruses-12-00697-f001:**
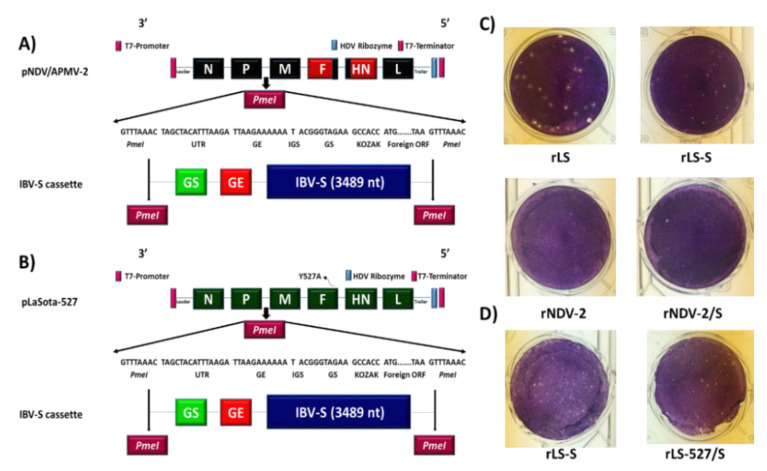
The schematic diagram for the construct and plaque morphology of rNDV-APMV-2, rLaSota-527 or rLaSota expressing the S protein of IBV. A plasmid BR322 containing cDNA of full genomic RNA of NDV strain Beaudette C (BC) in which its F and HN ectodomains were replaced with corresponded F and HN ectodomain of APMV-2 (pNDV/APMV-2) was constructed previously. A transcription cassette containing codon optimized S gene of the IBV strain Mass-41 was inserted between the P and M genes of NDV using *PmeI* restriction enzyme site (**A**). A plasmid BR322 containing cDNA of full genomic RNA of APMV-1 strain LaSota with Y527A mutation in cytoplasmic tail of its F gene (pLaSota-527) was constructed previously. The above transcription cassette containing codon optimized S gene of the IBV strain Mass-41 was inserted between the P and M genes of LaSota-527 using same strategy (**B**). The transcription cassette of the S gene contains the following sequences in a 3′ to 5′ constant order: *PmeI* restriction enzyme site, 15 nucleotides of NDV UTR, GE of NDV P gene, one T nucleotide IGS, GS of NDV M gene for S gene, Kozak, IBV S gene ORF, and *PmeI* restriction enzyme site. The plaque morphology of rNDV-APMV-2/IBV-S in comparison with plaque morphology of rLaSota/IBV-S in DF1 cells (**C**). The plaque morphology of rNDV-APMV-2/IBV-S and rLaSota-527/IBV-S in comparison with plaque morphology of rLaSota/IBV-S in DF1 cells (**D**).

**Figure 2 viruses-12-00697-f002:**
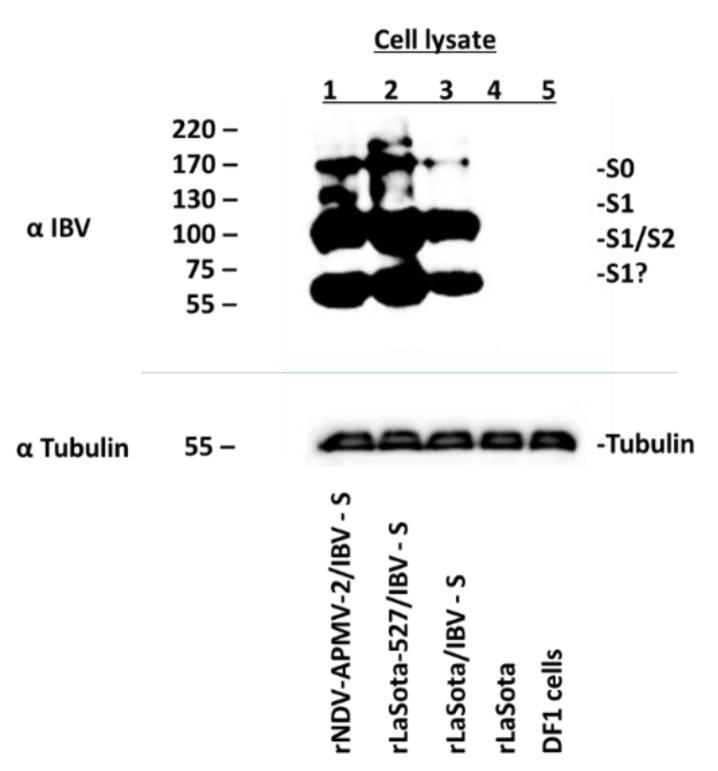
The expression of S protein of IBV by rNDV-APMV-2/IBV-S, rLaSota-527/IBV-S, and rLaSota/IBV-S. The expression of S protein by rNDV-APMV-2/IBV-S, rLaSota-527/IBV-S and rLaSota/IBV-S was detected in DF1 cells by Western blot analysis using a chicken polyclonal anti IBV serum (upper panel). Lanes 1, 2, 3, 4, and 5 represent rNDV-APMV-2/IBV-S, rLaSota-527/IBV-S, rLaSota/IBV-S, rLaSota, and DF1 cells, respectively. The expression of tubulin protein showing the loading of same amount of each cell lysate was detected in lower panel. Lanes 1, 2, 3, 4, and 5 represent rNDV-APMV-2/IBV-S, rLaSota-527/IBV-S, rLaSota/IBV-S, rLaSota, and DF1 cells, respectively.

**Figure 3 viruses-12-00697-f003:**
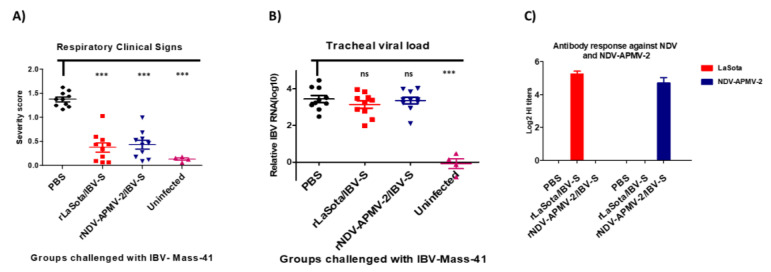
The protective efficacies of rNDV-APMV-2/IBV-S and rLaSota/IBV-S against IBV challenge in SPF chickens. Day-old chicks were immunized and four weeks after immunization, they were infected with IBV. The clinical signs of IBV were recorded twice a day for 10 days post-challenge. Each symbol represents mean scores of 10 days of clinical signs for individual birds and horizontal lines represent mean scores of each group, respectively (**A**). At day five post-challenge, tracheal swabs were collected and the IBV RNA load in each sample was quantified by RT-qPCR. Each symbol represents Log10 IBV RNA load for individual tracheal swab samples and horizontal lines represent mean RNA load of each group, respectively (**B**). The statistical difference between the PBS group and other groups was analyzed by *t*-test. ns, not significant (*p* > 0.05); *** significant (*p* < 0.05). Four weeks after immunization the sera were collected from all chickens and the antibodies induced against rNDV-APMV-2 and rLaSota were detected by HI assay. Serum titers are expressed as reciprocals Log2 dilution (**C**).

**Figure 4 viruses-12-00697-f004:**
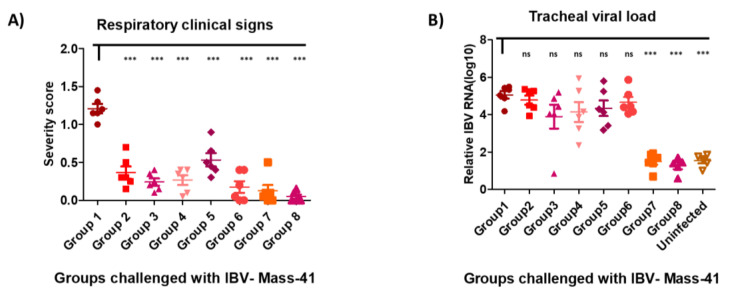
The protective efficacies of rNDV-APMV-2/IBV-S and rLaSota/IBV-S against IBV challenge in broilers. Chickens were immunized at day one and/or week three of age based on the groups listed in Table 1 and were infected with IBV at week six of age. The clinical signs of IBV were recorded twice a day for 10 days post-challenge. Each symbol represents mean scores of 10 days of clinical signs for individual birds and horizontal lines represent mean scores of each group, respectively (**A**). At day five post-challenge, tracheal swabs were taken and the IBV RNA load in each sample were quantified by RT-qPCR. Each symbol represents Log10 IBV RNA load for individual tracheal swab samples and horizontal lines represent mean RNA load of each group, respectively (**B**). The statistical difference between the Group 1 (PBS) and the other groups was analyzed by *t*-test. ns, not significant (*p >* 0.05); *** significant (*p <* 0.05).

**Figure 5 viruses-12-00697-f005:**
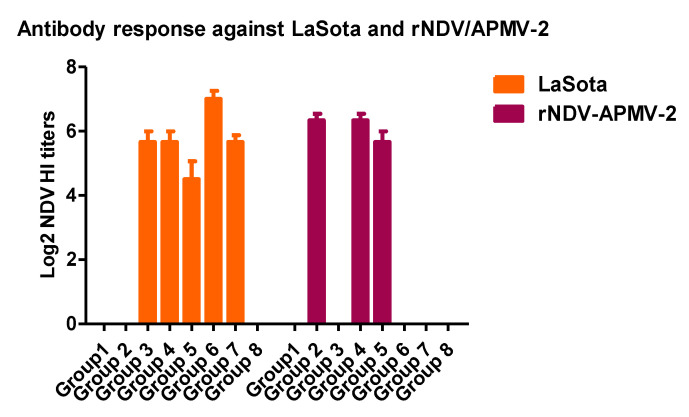
The antibodies induced against rNDV-APMV-2 and rLaSota in broiler chickens. Chickens were immunized at day one and/or week three of age based on the groups listed in Table 1. At week six of age, the sera were collected from all chickens and the antibodies induced against rNDV-APMV-2 and rLaSota were detected by the HI assay. Serum titers are expressed as reciprocals Log2 dilution.

**Table 1 viruses-12-00697-t001:** The groups of broilers immunized in single or prime-boost regimens.

Groups	Prime Immunization One-Day-Old	Boost-Immunization Three-Week-Old
1	PBS	PBS
2	rNDV-APMV-2/IBV-S	-
3	rLaSota/IBV-S	-
4	rNDV-APMV-2/IBV-S	rLaSota/IBV-S
5	rLaSota/IBV-S	rNDV-APMV-2/IBV-S
6	rLaSota/IBV-S	rLaSota/IBV-S
7	-	rLaSota/IBV-S
8	-	live IBV Vaccine

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
