# Peer review of "Comparative Protective Efficacies of Novel Avian Paramyxovirus-Vectored Vaccines against Virulent Infectious Bronchitis Virus in Chickens"

_viruses, 2020, doi:10.3390/v12070697_

Round 1
Reviewer 1 Report
The study describe the evaluation of a novel chimeric paramyxovirus serotype-2 (rNDV/APMV-2) expressing full S protein as vector vaccine against IBV and compare with the previously developed rLaSota vector ( rLaSota/IBV-S ) and modified version of rNDV strain LaSota (rLaSota-527) vector. Clinical signs but not virus shedding clearly reduced upon vaccination of one day old SPF chicks with rNDV/APMV-2 and rLaSota/IBV-S .Further, prime-boost immunization 'different compination' in in broiler cickens gave a comparable protection. However, single immunization rLaSota/IBV-S or live IBV vaccine, provided better protection against IBV. Further, the highly replicating rLaSota-527 vector expressing S protein provided less protection comparing to rLaSota/IBV-S. it is interesting study. However, there are some problems needed to pay attention.
- Abstract:
- Short description of the challenge studies required.
- Introduction:
- I think that the introduction part of the manuscript is not comprehensive enough, so the introduction part need be enhanced.
- It is not clear why using APMV2 safer comparing to APMV-1 strains as a vector vaccine system in chicken, this point should be highlighted in the introduction.
- Is there is difference between the receptor binding affinity, attachment and entrance of F and HN protein of APMV2 comparing to APMV-1 in avian cells? Can you add this information to the introduction?
- Lasota strain known to be highly replicating vaccine seed, is APMV2 has a comparable replication in embryonated chicken eggs? can you add this point to your introduction.
- Lines 86 to 94 are a repeat of line 23 to 29.
- Material and Methods:
- 111- 113: please add the name of plasmid used and accession number of F and HN used.
- 113-114: please add the accession number of APMV-1 strain LaSota with an Y527A previously published, or submit to gene bank.
- 115-117: please provide the method in detail for insertion of the intact S gene of IBV strain Mass-41 between P and M genes of NDV/APMV-2 and LaSota-527 or provide reference?
- 117-120: can you provide the primer sequences used for sequences confirmation or provide reference?
- 138-139: what is the MOI of each virus?
- 141: what is the source of the chicken polyclonal anti IBV serum?
- Results:
- Please provide data and/or references for the growth characterization and genetic stability of all constructs including in-ovo and/or in-vivo testing?
- Figure 1.A: what is pNDV/APMV-2 refer to?
- Figure 1.A: please explain what the black column at the end is and begin of F and HN gene respectively?
- Figure 1.A: what is pLaSota-527 refer to?
- Figure 1.C: There is difference in plaque morphology of rLS-S in the first and the 3rd raw despite they are the same construct or it is just due to low image quality, can you please provide better image quality?
- Table 1: please add the age/ day for prime and post immunization?
- Table 1: why there is no group for single immunization of rNDV/APMV-2 at 3 weeks?
- Figures : please show the significance difference between groups on all figures.
- Line 294: Is better protection here refer to clinical signs and/or virus shedding, can you please add the descriptive values on the text and add the reference of the previous work.
- Line 291-295: where is the antibody response against LaSota of rLaSota-527/IBV-S and rLaSota/IBV-S? This will give an idea about the impact of improving the replication.
- Why there is no neutralizing antibody response against IBV in all experiments?
- Discussion:
- Line 313 to 315: The used experimental approaches not fully support this claim. Please rephrase to descript manuscript findings.
- Line 313-330: There is no results for neutralizing antibody response against IBV in this work that make interpretation of the finding hard. Instead, Can you please discuss why the clinical signs clearly controlled while high viral titer comparable to positive control group still present and the correlation of local immunity from previous publications. Also, discuss the correlation between age of vaccine administration and intensity of immune response.
- Line 361-367: how you can confirm whether this protection solely due to vaccine or maturation of immune system including innate immunity.
- Line 370: How to confirm that the rLaSota-527/IBV-S more efficient replicating in-vivo comparing to rLaSota/IBV-S and whether this superiority in replication associated with higher immune response?
Author Response
Dear Reviewer,
Thank you for taking your time on our manuscript. I am pleased to submit the revised manuscript entitled,” Comparative protective efficacies of novel avian paramyxovirus-vectored vaccines against virulent infectious bronchitis virus in chickens.” Prepared by Edris Shirvani and Siba K. Samal for consideration as part of the Viruses Journal in the Animal Viruses, Avian Respiratory Viruses section. I am the corresponding author. We submitted the revised manuscript file and responses to peer-reviewers.
Please let us know if you have any questions.
Thank you.
Best Regards
Siba K. Samal
Professor of Virology
Virginia-Maryland College of
Veterinary Medicine – University of Maryland

Reviewer 2 Report
The manuscript represents substantial amount of research work using novel approach in designing and evaluating vectored vaccine against avian infectious bronchitis. Nevertheless, there are some issues that should be improved.
The rLaSota-527 vectored infcetious bronchitis virus (IBV) vaccine was used in this study solely for growth characterization and expression of the S protein assessment and not for protective efficacy evaluation in chickens as it was case for the other two vectored vaccines used in this study, rLaSota and rNDV/APMV-2 vectored IBV vaccines. This would bring confusion to readers, particularly in the abstract. It is stated later at the end of the results section of the manuscript that the results with the rLaSota-527 vectored IBV are not shown, because parts of data showing in the figures were presented in the author’s previous publication (Shirvani et al Sci Rep. 2018, 10, 8.) . Nevertheless, such results could not be found in the suggested article.
The authors stated in the introduction that rNDV/APMV-2 could be a potential viral vector for in ovo administration i.e. vaccination. In spite the fact that in ovo vaccination was not studied in this research, the authors claim at the beginning of the discussion that “this highly safe vector can be applied for in ovo vaccination”. Finally, at the end of the discussion the authors conclude that rNDV/APMV-2 vector expressing S protein is safe for in ovo vaccination without any evidence for the statement. Accordingly, the statement in the abstract “The rNDV/APMV-2 vector is much safer than rNDV vector, especially for in ovo immunization.” is not appropriate.
IBV exists in numerous serotypes with poor or no cross protection if the vaccine and the field virus are heterologous. In this study the protective efficacy of Massachusetts serotype vaccines (Baudette strain based) against homologous serotype challenge virus (Mass-41 strain) are evaluated which is proper approach, but this was not mentioned/elaborated in the abstract or in the rest of the manuscript. Even more, it is not clear what type of live attenuated IBV vaccine (homologous or heterologous) was used for the efficacy comparison with the vectored IBV vaccines.
The statement that the rNDV/APMV-2 vector will not be inhibited by NDV maternal antibodies (rows 311-312) should be an assumption since this was not confirmed experimentally. The importance of this possible advantage of the rNDV/APMV-2 vectored vaccine is likely marginal because inhibition of NDV based vectors by NDV maternal antibodies is negligible, particularly if the vaccine is administered at the age of three weeks as suggested by the authors.
Author Response

(The authors gave the same response as above.)
